# Somatostatin-Expressing Neurons Regulate Sleep Deprivation and Recovery

**DOI:** 10.3390/genes17010051

**Published:** 2026-01-01

**Authors:** Kenta Kobayashi, Y-h. Taguchi

**Affiliations:** 1Department of Physics, Graduate School of Science and Engineering, Chuo University, 1-13-27 Kasuga, Bunkyo-ku, Tokyo 112-8551, Japan; vv8vv8kenta@gmail.com; 2Department of Physics, Chuo University, 1-13-27 Kasuga, Bunkyo-ku, Tokyo 112-8551, Japan

**Keywords:** somatostatin, sleep deprivation, tensor decomposition, sleep debt

## Abstract

**Background/Objectives** We re-analyzed publicly available gene expression profiles from the male mouse cortex under conditions of sleep deprivation (SD) using tensor decomposition-based unsupervised feature extraction, originally proposed by one of the authors in 2017. **Methods** We focused on two distinct expression patterns: genes whose levels were altered in SD and failed to normalize during recovery sleep (RS), and genes that overshot normal levels during RS. This selection excluded the expected “altered in SD and recovered in RS” pattern, which was not significantly observed. These two gene sets showed substantial overlap but were still distinct from each other. **Results** The analysis revealed that the selected gene sets were enriched in various brain regions as evidenced through clustering in the Allen Brain Atlas. This suggests that the successful selection identified biologically meaningful genes. Furthermore, somatostatin (Sst)-expressing neuronal clusters were among the most highly enriched. **Conclusions** Given that sst is already implicated in SD and RS, our fully data-driven transcriptomic analysis successfully identified the activity of sst during SD and RS. These findings reveal that Sst-expressing neurons may play a key role in SD. These results were further validated using AlphaGenome by uploading the selected genes to it.

## 1. Introduction

Sleep deprivation (SD) is believed to increase sleep needs, which is met by subsequent recovery sleep (RS). Popescu et al. [1] re-analyzed gene expression profiles and observed that various pathways exhibited the expected pattern, implying that gene expression was altered during SD and returned to baseline following RS.

Although they hypothesized that RS alleviates the sleep pressure induced by SD, they asserted that this hypothesis-driven approach was limited in its ability to fully explore large datasets, as it may overlook unexpected findings. In contrast, a data-driven approach has the potential to accurately determine the underlying process between sleep deprivation and sleep recovery.

In this study, we aimed to re-analyze the dataset from Popescu et al. using an entirely data-driven approach: tensor decomposition (TD)-based unsupervised feature extraction (FE) [2]. We observed two patterns of dependence: home cage (control) vs. sleep deprivation vs. recovery sleep. These patterns were expressed as singular value vectors obtained using TD. Genes whose expression correlated with these two patterns were identified and observed to be enriched in various biological processes, neuronal categories (based on the Allen Brain Atlas), miRNA target genes, and histone modifications. Specifically, the gene sets were strongly associated with the somatostatin (Sst) cell cluster in the Allen Brain Atlas, which is related to sleep regulation.

During sleep, Sst neurons are involved in disrupting sleep and impairing cognitive function under certain circumstances. This dual characteristic indicates that the function of Sst neurons is strongly influenced by the brain region and condition.

One example is its role in the hippocampus during SD. SD has been shown to selectively activate Sst interneurons in the hippocampus. This activation acts as a “gate” that inhibits activity in the dentate gyrus, which is essential for memory fixation and interferes with synaptic plasticity and information transmission. Experimentally, simply mimicking the activation of Sst neurons inhibits memory fixation, even in animals with a normal sleep pattern. Conversely, inhibiting these neurons during post-learning sleep promotes memory fixation [3]. This suggests that Sst neurons are part of the direct mechanism by which SD induces cognitive impairment.

Another important example is the circuit in the central nucleus of the amygdala (CeA) that links stress and sleep disturbances. Here Sst neurons (CeA^*SST+*^) in the CeA are activated by acute stress [4]. The optogenetic activation of these neurons induces a rapid transition from NREM sleep to wakefulness, prolonging the time required to fall asleep (onset latency) [4]. Critically, the inhibition of the activity of these neurons ameliorates stress-induced insomnia but has no effect on normal physiological sleep [4]. This shows that CeA^*SST+*^ neurons are not part of the normal sleep–wake cycle but are the basis of pathological sleep disturbances resulting from specific stress conditions.

These findings show that the function of Sst neurons changes dramatically from “sleep promoting” to “sleep inhibiting” depending on the brain region (cortex, hippocampus, amygdala) and brain state (normal sleep, SD, stress). This is not a contradiction but evidence of sophisticated context-dependent circuit regulation, showing that different systems (homeostasis, stress response, and memory) mobilize the powerful network regulators of Sst neurons for different and sometimes conflicting purposes.

The effects of Sst neurons on sleep are tightly regulated by neural circuits between specific brain regions (Table 1). Of particular importance are the top–down control from the prefrontal cortex to the hypothalamus and structural changes caused by sleep deprivation in the hippocampal and neocortical circuits.

The prefrontal cortex (PFC) is responsible for higher executive functions and is highly sensitive to sleep deprivation. Recent studies have revealed the existence of a top–down control mechanism for sleep homeostasis, in which the PFC monitors the internal state of sleep debt and directs appropriate physiological and behavioral responses [5].

Activity-dependent gene labeling techniques using c-Fos have identified a specific subpopulation of PFC Sst-GABAergic neurons that are particularly excitatory and activated in response to sleep deprivation [5,6]. These activated PFC Sst neurons have long-range projections to two distinct nuclei in the hypothalamus, each of which has been shown to be responsible for a distinct function by optogenetic manipulation.

Projection to the lateral preoptic area (LPO): Selective stimulation of the axon terminals of these PFC Sst neurons with the LPO elicited sleep preparation behaviors, such as nesting behavior. This behavior is accompanied by physiological changes, such as theta wave enhancement and increased body temperature but does not induce sleep itself [5]. The LPO is part of the preoptic area, a long-established sleep facilitation center [7] that provides a new mechanism through which the cortex communicates with the sleep center and initiates preparatory behaviors before sleep. This offers a new mechanism by which the cortex communicates with the sleep center to promote preparatory behaviors before sleep.

Projection to the lateral hypothalamus (LH): In contrast, stimulation of the axon terminals of the same group of neurons in the LH does not induce nesting behavior but instead triggers robust NREM sleep (recovery sleep) with strong delta waves, accompanied by a decrease in body temperature [5].

This circuit highlights a sophisticated top–down control, in which the PFC senses sleep debt and commands the hypothalamus via parallel Sst neuronal pathways to “first prepare a safe place to sleep (via LPO) and then initiate restorative sleep (via LH)”, a two-stage coordinated homeostatic response. This challenges the classical view that sleep is exclusively driven by subcortical, bottom–up mechanisms.

The effects of sleep deprivation go beyond functionality and leave physical traces on Sst interneuron circuits. Using state-of-the-art imaging techniques, studies have reported that sleep deprivation for only a few hours can induce region-specific structural changes in the dendritic spines (postsynaptic structures) of Sst interneurons in the hippocampus and neocortex [8]. These changes were not uniform; for example, there was a dramatic increase in the density and size of spines in the CA3 region of the hippocampus and a slight decrease in the visual cortex.

Considering dendritic spines are key sites of excitatory input, this rapid and widespread structural change indicates that sleep deprivation fundamentally alters excitatory input to Sst interneurons. This disrupts the excitatory (E-I) balance across critical neural networks through changes in the inhibitory output from these neurons [8]. This finding lays out a physical mechanism at the cellular level underlying the cognitive decline and changes in network activity observed after sleep deprivation. This is a process by which the brain’s inhibitory circuits are dynamically and physically sculpted by sleep, and whose disruption is the core mechanism of sleep deprivation-induced dysfunction.

**Table 1 genes-17-00051-t001:** Region-specific roles of Sst neurons in sleep–wake regulation. ↑: increased.

Brain Region	Context/Trigger	Function of Sst Neurons	Effect on Sleep/Wakefulness	Refs
Neocortex	Normal Sleep	Generates “OFF state” of slow waves	SWA ↑, NREM sleep time ↑	[9]
Hippocampus	Sleep Deprivation	Inhibitory gate for memory consolidation	Inhibits sleep function (memory)	[3]
Central Amygdala (CeA)	Acute Stress	Mediates stress-induced arousal	Sleep latency ↑, Arousal ↑	[4]
Prefrontal Cortex (PFC) Projection	Sleep Deprivation (Homeostatic Response)	Coordinates recovery sleep	Induces preparatory behavior and recovery NREM sleep	[5]

## 2. Materials and Methods

Figure 1 shows the flow chart representing analysis workflow in this study. For a more detailed explanation of the statistical methods used in this study, please see [2] and the numerous papers listed in its publicly available Appendix B [2].

### 2.1. Gene Expression Profiles

The gene expression profiles analyzed in this study were sourced from Gene Expression Omnibus (GEO) using the GEO ID GSE237419 [1]. The file GSE237419_gse_HCSDRS _WT_SleepIntegration_salmon.txt was retrieved and loaded into R using the read.csv command. Once the dataset was loaded to R, it did not undergo any further treatment. Therefore, it is suggested that the original paper is referred for any data treatments.

### 2.2. Tensor

Gene expression profiles were formatted as a tensor $x_{ijkm}\in R^{N\times 3\times 5\times 5}$ which represents the expression of the *i*th gene of the *j*th condition (j=1: home cage (HC), j=2: sleep deprivation (SD), and j=3: recovery sleep (RS)) of the *k*th replicate at the *m*th hour. (m=1: three hours, m=2: five hours, m=3: six hours, m=4: seven hours, and m=5: 12 h), where N=54,308 denotes the total number of genes. As not all combinations of i,j,k, and *m* were measured, the missing values were filled with zeros (Table 2).

### 2.3. Tensor Decomposition

Before applying TD to the obtained tensor, $x_{ijkm}$ and $x_{ijkm}$ were standardized as follows:(1)$$\begin{array}{ccc} \sum_{i=1}^{N}x_{ijkm} & = & 0 \end{array}$$(2)$$\begin{array}{ccc} \sum_{i=1}^{N}x_{ijkm}^{2} & = & N \end{array}$$

After applying HOSVD [2] to $x_{ijkm}$, we get(3)$$x_{ijkm}=\sum_{\ell_{1}=1}^{N}\sum_{\ell_{2}=1}^{3}\sum_{\ell_{3}=1}^{5}\sum_{\ell_{4}=1}^{5}G\left(\ell_{1}\ell_{2}\ell_{3}\ell_{4}\right)u_{\ell_{1}i}u_{\ell_{2}j}u_{\ell_{3}k}u_{\ell_{4}m}$$
where $G\in R^{N\times 3\times 3\times 5}$ denotes the core tensor representing the contribution of the $u_{\ell_{1}i}u_{\ell_{2}j}u_{\ell_{3}k}u_{\ell_{3}m}$ to $x_{ijkm}$, $u_{\ell_{1}i}\in R^{N\times N}$, $u_{\ell_{2}j}\in R^{3\times 3}$, $u_{\ell_{3}k}\in R^{5\times 5}$, and $u_{\ell_{4}m}\in R^{5\times 5}$ are the singular value matrices, and orthogonal matrices.

### 2.4. Gene Selection

Although the procedure was described in detail in a recent book [2], it is given here briefly. To select $u_{\ell_{1}i}$ for gene selection, we need to identify which $u_{\ell_{2}j}$, $u_{\ell_{3}k}$, and $u_{\ell_{4}m}$ are of interest as follows:Since the gene expression ought to be distinct between controls and others, $u_{\ell_{2}1}$ should be distinct from $u_{\ell_{2}2}$ and/or $u_{\ell_{2}3}$.Since the gene expression should not be distinct between replicates, $u_{\ell_{3}k}$ should be constant as well as possible.Since we did not pre-judge dependence upon time, we overlooked the dependence of $u_{\ell_{4}m}$ upon *m*.

After identifying $\ell_{2}$, $\ell_{3}$, and $\ell_{4}$ as items of interest, we investigated $G(\ell_{1}\ell_{2}\ell_{3}\ell_{4})$ by fixing $\ell_{2}$. $\ell_{3}$ and $\ell_{4}$ to the selected ones, and selected $\ell_{1}$ whose associated *G* demonstrated the largest absolute value.

*P* values were assigned to *i*s after assuming the null hypothesis that $u_{\ell_{1}i}$ is Gaussian following the equation(4)$$P_{i}=P_{\chi^{2}}\left[>{\left(\frac{u_{\ell_{1}i}}{\sigma_{\ell_{1}}}\right)}^{2}\right]$$
where $P_{\chi^{2}}\left[>x\right]$ is the cumulative Gaussian distribution, where the argument is larger than *x*, and $\sigma_{\ell_{1}}$, which is optimized (see below), is the standard deviation of $u_{\ell_{1}i}$. Finally, genes associated with *P*-values adjusted by the BH criterion [2] to less than 0.01 were selected.

The motivation behind optimizing $\sigma_{\ell_{1}}$ was as follows: If we compute $\sigma_{\ell_{1}}$ using all $u_{\ell_{1}i}$ including those attributed to *i*s to be selected as outliers, $\sigma_{\ell_{1}}$ is likely to be overestimated, and $P_{i}$ might be overestimated. Using the overestimated $P_{i}$, we identified fewer *i*s associated with significant $P_{i}$s. Thus, it is important to estimate $\sigma_{\ell_{1}}$ excluding *i*, which is to be selected as an outlier.

Suppose that $h_{s}\left(1-P_{i}\right)$ is the histogram of the *s*th bin (1≤s≤S) of $1-P_{i}$ computed by substituting the arbitrary value of $\sigma_{\ell_{1}}$ into Equation (Equation 4), excluding *i* values associated with *P*-values adjusted by a BH criterion of less than 0.01. We compute the standard deviation of $h_{s}$, $\sigma_{h_{s}}$ as follows:(5)$$\begin{array}{ccc} \langle h_{s}\rangle & = & \frac{1}{S}\sum_{s=1}^{S}h_{s} \end{array}$$(6)$$\begin{array}{ccc} \sigma_{h_{s}} & = & \sqrt{\frac{1}{S}\sum_{s=1}^{S}{\left(h_{s}-\langle h_{s}\rangle \right)}^{2}} \end{array}$$

### 2.5. AlphaGenome

We summarize how we retrieved features from the AlphaGenome.

#### 2.5.1. Conversion to Genomic Regions

Since AlphaGenome does not accept the list of gene names/IDs, the Ensembl gene IDs of the selected genes were converted to genomic regions using BSgenome.Mmusculus. UCSC.mm39 [10].

#### 2.5.2. Upload to AlphaGenome and Retrieving Features

The genomic regions obtained were uploaded to AlphaGenome using the Python (Ver. 0.2.0) API [11]. The DNase-Seq prediction by AlphaGenome of uploaded regions was performed using the Python API.

#### 2.5.3. The Selection of Top Ranked Genes

The brain regions considered were “Brain”, “left cerebral cortex”, “right cerebral cortex”, “layer of hippocampus”, and “frontal cortex”. Comparisons were performed within four pairwise comparisons, i.e., “Brain” vs. “left cerebral cortex”, “Brain” vs. “right cerebral cortex”, “Brain” vs. “layer of hippocampus”, and “Brain” vs. “frontal cortex”. Logarithmic ratios were computed ($10^{-6}$ was added to the denominator to avoid division by zero), and the top 100 genes with the larger/smaller logarithmic ratios were selected. The 100 top ranked genes were uploaded to Enrichr.

### 2.6. Alternative Gene Expression Profiles

We employed the following three gene expression profiles for the validation, although their data structures are not fully identical to that of GSE237419.

#### 2.6.1. GSE33491 [12]

Nine CEL files, from GSM828577.CEL.gz to GSM828585.CEL.gz were downloaded and normalized by RMA. It was formatted as a tensor, $x_{ijk}\in R^{N\times 3\times 3}$, which represents the gene expression profile of the *i*th gene of the *j*th condition (j=1 HC, j=2 SD, and j=3 RS) of the *k*th replicate.

#### 2.6.2. GSE78215 [13]

We downloaded the six CEL files whose file names started with “Cortex_Circadian Control_ZT0” as HC, with “Cortex_6HrSleepDep_ZT6” as SD, and with “Cortex_ 6HrSDw1HrRecovery_ZT7” as RS. After being normalized using RMA, they were formatted as a tensor, $x_{ijk}\in R^{N\times 3\times 6}$, which represents the gene expression profile of the *i*th gene of the *j*th condition (j=1 HC, j=2 SD, and j=3 RS) of the *k*th replicate.

#### 2.6.3. GSE144957 [14]

The file named as GSE144957_SLEEPSEQ_CPM_ALL.txt.gz was downloaded. Afterward, six WT_CS samples, from GSM4301443 to GSM4301448, were used as HC, six WT_SD samples, from GSM4301449 to GSM4301454, and six WT_RS samples, from GSM4301456 to GSM4301461, were downloaded. They were formatted as a tensor, $x_{ijk}\in R^{N\times 3\times 6}$, which represents the gene expression profile of the *i*th gene of the *j*th condition (j=1 HC, j=2 SD, and j=3 RS) of the *k*th replicate.

## 3. Results

### 3.1. Gene Selection

First, we investigated $u_{\ell_{2}j}$ to determine the type of dependence among the HC, SD, and RS conditions (Figure 2). Contrary to the expectation, no simple “altered in SD and recovered in RS” pattern was observed but “altered in SD and not recovered in RS” ($u_{2j}$) and “altered in SD and overshot in RS” ($u_{3j}$) patterns were observed. However, $u_{1k}$ (Figure 3, left panel) and $u_{1m}$ (Figure 3, right panel) do not express dependence on *k* and *m*. Thus, we sought the largest $G(\ell_{1},2,1,1)$ (Figure 4, left panel) and $G(\ell_{1},3,1,1)$ (Figure 4, right panel) values. We decided to use $\ell_{1}=2,3$ for the gene selection.

Next, we attempted to optimize $\sigma_{\ell_{1}}$ for $\ell_{1}=2,3$ (see Figure 5). As expected, we successfully optimized $\sigma_{\ell_{1}}$ to minimize $\sigma_{h_{s}}$. $P_{i}$ was computed using Equation (Equation 4) and adjusted using the BH criterion [2]. As a result, 2182 for $\ell_{1}=2$ and 2792 for $\ell_{1}=3$ ensemble genes were identified to be associated with the corrected *p*-values less than 0.01. The ensemble genes were converted to 2081 (for $\ell_{1}=2$) and 2089 (for $\ell_{1}=3$) gene symbols. Appendix A shows a Venn diagram of the gene symbols selected by $\ell_{1}=2$ and $\ell_{1}=3$.

Despite the significant overlap, these two remain distinct.

### 3.2. Metascape

The gene symbols were uploaded to Metascape [15] to biologically evaluate the selected genes. Table 3 lists the top 20 clusters. These include terms such as “central nervous system neuron axonogenesis”, which is consistent with the fact that the samples were obtained from brain tissue, and terms like “autophagy” and “metabolism”, which were highlighted in the analysis by Popescu et al. [15]. Furthermore, endoplasmic reticulum stress is known to be induced by sleep loss [16] (“GO:0034976, response to endoplasmic reticulum stress”), sleep loss can alter the structures of various organelles, including mitochondria, nucleus, and Golgi apparatus [17] (“GO:0048193, Golgi vesicle transport”). Additionally, locomotory behavior was identified, which aligns with previous studies discussing motor control mechanisms during sleep [18] (“GO:0007626, locomotory behavior”). A full enrichment analysis, shown in Appendix A, reveals further overlap with Popescu et al.’s findings, including “Wnt signaling pathway”, “Notch signaling pathway”, and “Mapk signaling pathway”. These results indicate that the genes identified are biologically functional.

To further validate the biological relevance of the identified genes, we examined the protein–protein interaction network and MCODE components generated from the gene lists (Appendix A and Table 4) using Metascape. These results appear biologically plausible, as they include “MTOR signaling” and “Pathways in cancer”, which are reported in the study by Popescu et al. [15] and multiple Rho GTPases related terms, which were addressed in our previous work [19], were detected.

### 3.3. Enrichr

We uploaded the selected genes to Enrichr [20] and observed that many enriched cell clusters exist in the “Allen Brain Atlas 10x scRNA 2021” category (Figure 6 and Appendix A).

We observed that the cell cluster labeled as “Sst”, referring to “MGE-derived GABAergic subclass typically expressing Sst”, was highly enriched in $\ell_{1}=2$ and $\ell_{1}=3$.

Gabaergic Sst may play a critical role in SD and RS. All brain functions depend on a delicate balance between excitatory and inhibitory signals [21]. In the mature central nervous system, amma-aminobutyric acid (GABA) is the main inhibitory neurotransmitter, playing a vital role in preventing the excessive excitation of neural networks and regulating information transmission [22]. Although GABAergic interneurons account for only approximately 10–20% of all neurons, their diversity and unique connectivity significantly enhance cortical computations and information processing [23]. Dysfunctions in these interneurons are associated with several neuropsychiatric disorders, including epilepsy, schizophrenia, and anxiety.

Among GABAergic interneurons, Sst-expressing neurons form a major class, comprising approximately 30% of cortical interneurons [24]. These neurons co-release the GABA and Sst, potentially enhancing GABAergic inhibitory effects [25]. Anatomically and electrophysiologically, these cells are heterogeneous, including Martinotti cells and low-threshold spiking cells [26]. Sst neurons are widely distributed in brain regions such as the neocortex and hippocampus, which are important for higher cognitive functions and memory [27].

Research on Sst neurons revealed two seemingly contradictory findings. On the one hand, these neurons play an essential role in generating electroencephalogram (EEG) slow-wave oscillations, characteristic of slow-wave sleep, which contributes to deep and restorative sleep [9]. On the other hand, they are crucial for generating brain waves (EEG), a process that is known as “slow-wave sleep”. In this context, Sst neurons function as “sleep builders”. At the same time, however, they have been evidenced to be deeply involved in processes that actively interfere with sleep, such as stress-induced insomnia and cognitive dysfunction due to sleep deprivation [3]. In the following section, we aim to answer this central paradox: How can the same types of neurons promote and disrupt sleep? The answer lies in the “context-dependence function” of Sst neurons, which is not determined by their neurotransmitters but by the specific neural circuits they are embedded in and the dynamic state of the brain. This report aims to systematically explain the complex relationship between Sst neurons and sleep regulation by exploring the contextual dependence.

Non-rapid eye movement (NREM) sleep, particularly in its deepest stage, slow-wave sleep, is essential for brain recovery, memory consolidation, and metabolite elimination [28]. The EEG of this sleep stage is characterized by low-frequency, high-amplitude “slow waves” below 1 Hz [9]. These slow waves are caused by a periodic cycle of “ON states” (periods of depolarization and firing) and “OFF states” (periods of hyperpolarization and silence), in which a large number of cortical neurons are synchronously active and cease to be active altogether [9].

Recent studies have shown that cortical Sst neurons play a central role in triggering synchronous OFF states. Using optogenetics, the brief activation of cortical Sst neurons in the ON state induces a long OFF state that persists beyond the stimulation time, thereby silencing a broad population of cortical neurons [9]. This effect is attributed to the ability of Sst neurons, particularly Martinotti cells, to project extensive axons onto cortical layer 1, where the distal dendrites of pyramidal cells are densely located, causing strong and sustained GABA_*B*_ receptor-mediated inhibition [24].

Furthermore, the sustained activation of Sst neurons using chemogenetics significantly increases slow wave activity (SWA), steepens the slope of individual slow waves, and lengthens the total duration of NREM sleep. Conversely, the inhibition of these neurons significantly reduces SWA [9]. This function is specific to Sst neurons. For example, the activation of another major interneuron, parvalbumin-expressing neurons, increases NREM sleep duration but significantly decreases SWA, causing an overall inhibition rather than a structured oscillation in cortical activity [29]. These findings suggest that Sst neurons do not merely facilitate sleep but act as “builders” of slow waves that determine restorative sleep quality.

In conclusion, it is reasonable to detect Sst expressive neurons that regulate SD and RS (Table 1).

### 3.4. Alternative Gene Expression Profiles

To confirm that the above mentioned findings are seen in the other datasets, we employed three alternative datasets. The expected “altered in SD and recovered in RS” were observed for one of $u_{\ell_{2}j}$s of all three profiles and corresponding $u_{\ell_{1}i}$ was used for gene selection. Nevertheless, the optimization of $\sigma_{\ell_{1}}$ was not successful, and the flat histogram of $P_{i}$ that is coincident with the null hypothesis that $u_{\ell_{1}i}$ obeys Gaussian was not observed (Figure 7). Consequently, the selected genes were inevitably very small. For GSE33491, only 140 probes were selected, corresponding to 28 gene symbols. For GSE78215, only 141 probes were selected, corresponding to six gene symbols. For GSE144957, no genes were selected. Since TD-based unsupervised FE failed to select a sufficient number of genes for these three profiles, we were unable to validate our results using these alternative gene expression profiles.

## 4. Discussion

To determine whether the detection of Sst enrichment was accidental and might not be supported when using other samples, we uploaded the two sets of selected genes to AlphaGenome [30] to identify if the Sst enrichment pattern persisted. AlphaGenome is a generative AI server that can predict various omics features (gene expression, DNA methylation, etc.) only from DNA sequences (for more details on how to obtain predicted features from AlphaGenome, see the Section 2).

Table 5 summarizes if Sst is in the top-ranked 10 cell types and/or associated with adjusted *P*-values less than 0.05 in “Allen Brain Atlas 10x scRNA 2021” (the full list of corresponding Appendix A is in Appendix A). Sst was almost always ranked in the top 10 cell types, and in 16 out of 80 cases, Sst showed adjusted *P* values less than 0.05. All eight pairwise comparisons were associated with at least one case with adjusted *P*-values less than 0.05. As DNase-Seq identified the transcriptionally active sites, which are expected to correspond to the actively expressed genes, the Sst enrichment observed in DNase-Seq predicted by AlphaGenome suggests that it is not accidental and is supported by other samples as well.

We intentionally did not fully compare our enrichment analyses with those in Popescu et al. [1], excluding partial discussions in Metascape section. Since Popescu et al. used unadjusted (raw) *P*-values for the statistical significance of enrichment analysis, it was not credible at all, whereas our enrichment analyses were statistically highly significant (Table 4 and Figure 6). Popescu et al.’s not statistically significant enrichment was not worthwhile compared fully with ours.

Readers may find the detailed description of the method incomplete. Although this method is specialized and not popular at all, it is a fully established method. The method has the origin in PCA-based unsupervised FE proposed in 2012 and was developed to TD-based unsupervised FE in 2017. We have published more than 70 peer-reviewed papers/conference proceedings using these methods as listed in Appendix B [2], which is publicly available. There are two editions of hard cover books that describe these methods, the latter of which is more than 500 pages long [2]. It is unrealistic to fully describe such a method that has a dynamic history every time the new paper is published. The readers who are deeply interested in the methods should refer to these books as well as more than 70 published papers. Thus, we believe that the outline included in this manuscript is enough for the majority of readers who are not particularly interested in methodological details but biological outcomes (e.g., enrichment analysis).

In addition to this, substituting zeros to missing values might appear problematic. Nevertheless, this strategy was proven to work well [2]. For example, in multiomics data where more than 90 % are missing, substituting the missing entries with zeros is a common strategy [31]. Thus, substituting zeros for missing values can perform well for tensor decomposition.

Optimizing $\sigma_{\ell_{1}}$ might sound questionable, given the relatively large number of the selected genes (∼2000). Nonetheless, according to the previous study [32], the optimization of $\sigma_{\ell_{1}}$ can improve the biological significance of the selected genes even when a greater number of genes are selected.

One may also question the relevance of TD; simpler methods, e.g., PCA, might achieve the same results. To refute this possibility, we applied PCA to the unfolded matrix, $x_{i(jkm)}\in R^{N\times 42}$, where 42 is the total number of samples (Figure 8). Since it is obvious that there are no interpretable patterns, TD is clearly superior to PCA.

## 5. Conclusions

In this study, we analyzed time-dependent gene expression in the prefrontal cortex of mice during SD and RS. Compared with control, we identified two distinct expression patterns using TD, “altered during SD but not recovered during RS” and “altered during SD with an overshoot during RS”, which differ from the commonly expected pattern “altered during SD and recovered during RS”. We extracted two sets of genes associated with these patterns and performed enrichment analysis, which revealed enrichment in Sst-expressing cell types. To confirm that the Sst enrichment was not due to chance, we uploaded both gene sets to AlphaGenome and retrieved predicted DNase-Seq. The top-ranked genes distinguishing various brain subregions from the whole brain were enriched in Sst cell types. This suggests that Sst enrichment is not accidental, even when other samples are used.

To our knowledge, this is the first study to demonstrate that a fully data-driven transcriptomic analysis can successfully identify the function of sst during SD and RS. This finding suggests that this fully data-driven approach is a valuable method for uncovering diverse genetic functions underlying sleep function. We anticipate that this methodology will be widely applicable to studying various aspects of sleep mechanisms from a genetic perspective.

## Figures and Tables

**Figure 1 genes-17-00051-f001:**
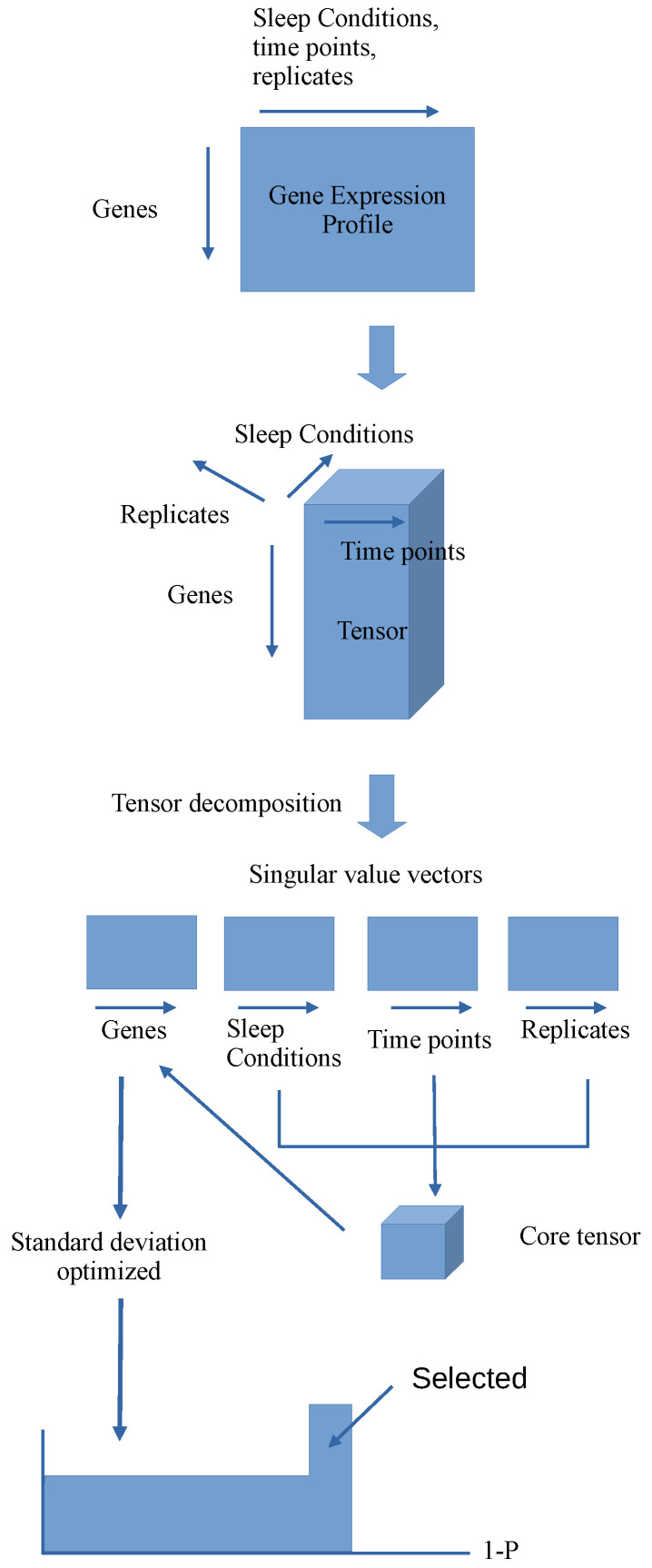
Flow chart representing analysis workflow. The study began with the gene expression matrix, whose row and column are attributed to the genes and the mixture of sleep conditions, time points and replicates, respectively. Tensor decomposition was then applied to the tensor generated from the gene expression matrix. The singular value vectors attributed to sleep conditions, time points, and replicates were first investigated and selected. The singular value vector attributed to genes were selected through core tensor compared with the selected singular value vectors attributed to sleep conditions, time points, and replicates. *p*-values were calculated for the genes factor scores, assuming the null hypothesis that the singular value vector attributed to genes obeys Gaussian, with optimized standard deviation. Finally, genes associated with the adjusted *p*-values less than 0.01 were selected for further analysis (e.g., enrichment analysis).

**Figure 2 genes-17-00051-f002:**
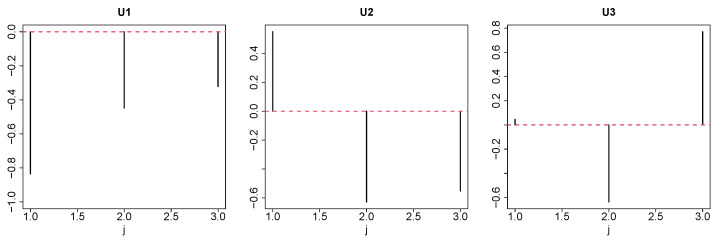
From left to right: $u_{1j}$, $u_{2j}$, and $u_{3j}$. j=1: HC, control, j=2: SD, and j=3: RS. Red broken lines represent baselines.

**Figure 3 genes-17-00051-f003:**
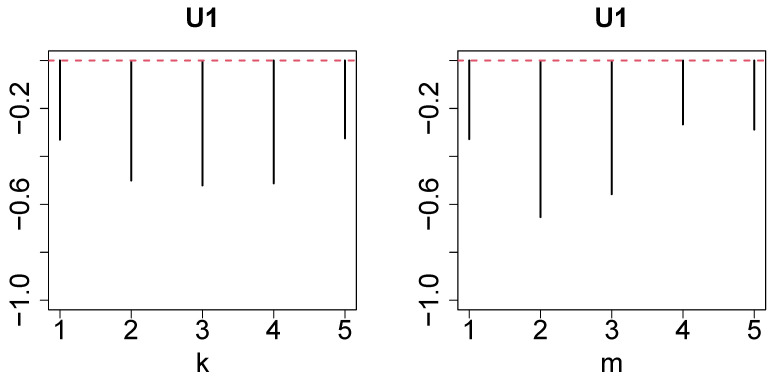
**Left**: $u_{1k}$ and **right**
$u_{1m}$. Red broken lines denote baselines.

**Figure 4 genes-17-00051-f004:**
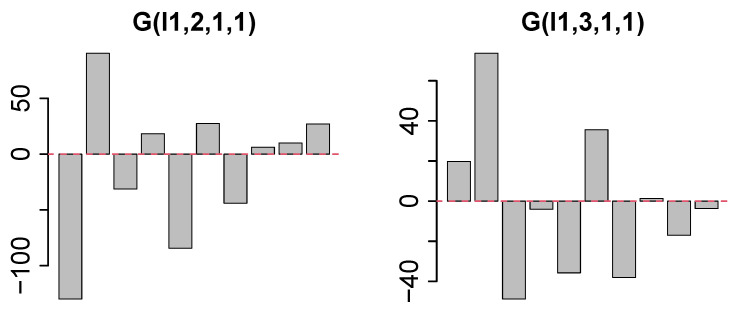
**Left**: $G(\ell_{1},2,1,1)$ and **right**
$G(\ell_{1},3,1,1)$. Horizontal axis is $1\leq \ell_{1}\leq 10$ from left to right.

**Figure 5 genes-17-00051-f005:**
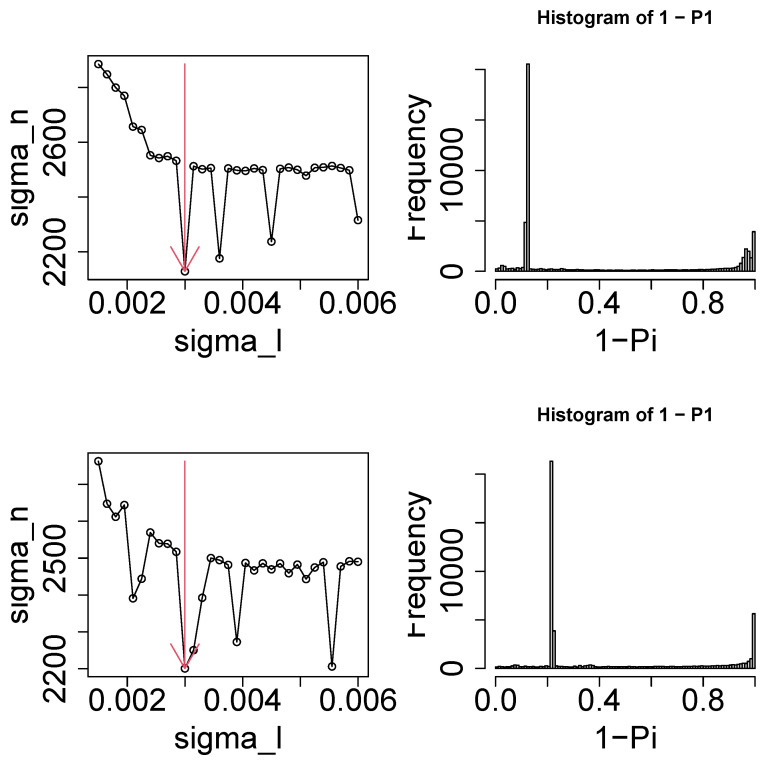
**Left column**: $\sigma_{h_{s}}$ as a function of $\sigma_{\ell_{1}}$. Vertical red arrows suggest the selected $\sigma_{\ell_{1}}$. **Right column**: The histogram of $1-P_{i}$ computed by Equation (Equation 4) using the selected $\sigma_{\ell_{1}}$. **Upper**: $\ell_{1}=2$. **Lower**: $\ell_{1}=3$.

**Figure 6 genes-17-00051-f006:**
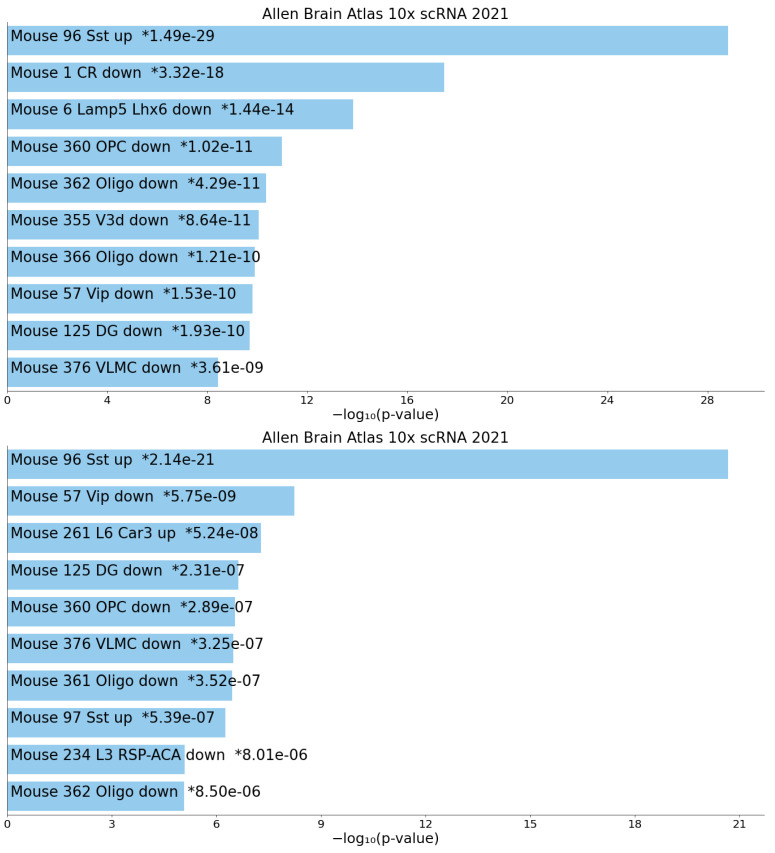
The bar chart shows the top 10 enriched terms in the chosen library, along with their corresponding *p*-values. Colored bars correspond to terms with significant *p*-values (<0.05). An asterisk (*) next to a *p*-value indicates the term also has a significant adjusted *p*-value (<0.05). **Upper**: $\ell_{1}=2$ and **lower**: $\ell_{1}=3$.

**Figure 7 genes-17-00051-f007:**
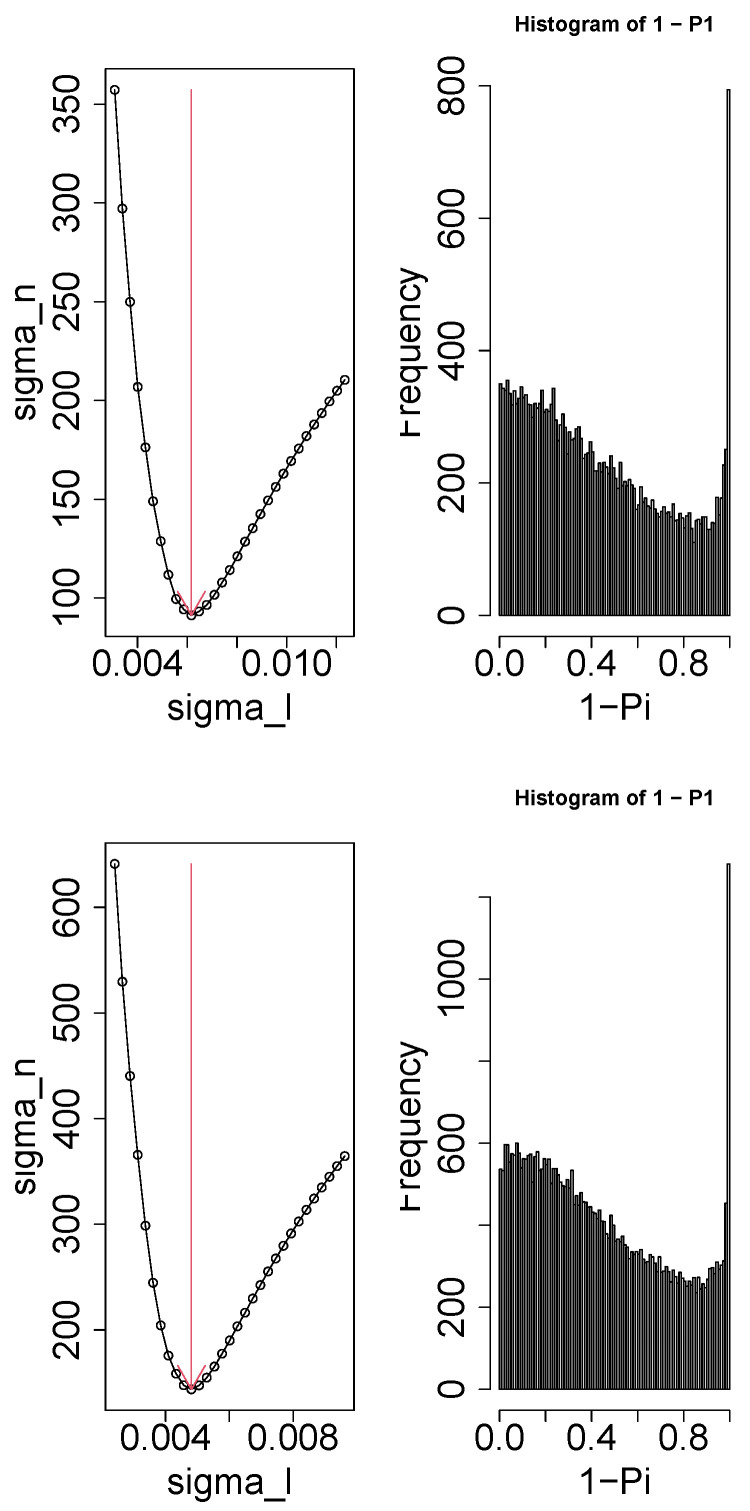
Left column: $\sigma_{h}$ as a function of $\sigma_{\ell_{1}}$. Vertical red arrows suggest the selected $\sigma_{\ell_{1}}$. Right column: The histogram of $1-P_{i}$ computed by Equation (Equation 4) using the selected $\sigma_{\ell_{1}}$. **Top row**: GSE33491, **middle row**: GSE78215, and **bottom row**: GSE144957.

**Figure 8 genes-17-00051-f008:**
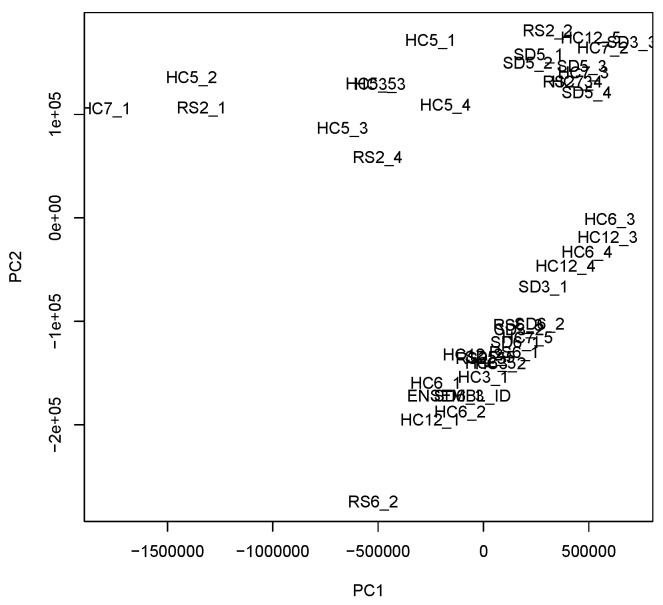
The results of PCA applied to $x_{i(jkm)}$.

**Table 2 genes-17-00051-t002:** Correspondence between indices, j,k, and *m*, of the tensor, $x_{ijkm}$, and experimental conditions. When the number of replicates is not five but three, only $x_{ijkm}$ with 2≤k≤4 have values. $x_{ijkm}$s whose (j,k.m)s are not listed were filled with zero. HC: Home cage (control), SD: Sleep deprivation, and RS: Recovery sleep. The numbers adjusted to two alphabets, HC, SD, and RS, represent hours. For more details, see [1].

m	j=1	j=2	j=3
1	HC3 (2≤k≤4)	SD3 (2≤k≤4)	—
2	HC5 (1≤k≤5)	SD5 (1≤k≤5)	RS2 (1≤k≤5)
3	HC6 (1≤k≤5)	SD6 (2≤k≤4)	RS6 (2≤k≤4)
4	HC7 (1≤k≤5)	—	—
5	HC12 (1≤k≤5)	—	—

**Table 3 genes-17-00051-t003:** Top 20 clusters and their representative enriched terms (one per cluster) identified using Metascape, based on 2081 (for $\ell_{1}=2$) and 2089 (for $\ell_{1}=3$) gene symbols uploaded. “Count” refers to the number of user-provided genes associated with a given ontology term. “%” indicates the percentage of all of the user-provided genes that are found in the given ontology term (only input genes with at least one ontology term annotation are included in the calculation). “Log10(P)” is the *p*-value in log base 10. “Log10(q)” is the multi-test adjusted *p*-value in log base 10. PC2: $\ell_{1}=2$ and PC3: $\ell_{1}=3$.

PC2	PC3	GO/REAC	Category	Description	Count	%	Log10(P)	Log10(q)
	◯	R-MMU-72766	REAC	Translation	65	4.16	−21.75	−17.49
◯	◯	R-MMU-8953854	REAC	Metabolism of RNA	112	7.17	−21.43	−17.47
◯	◯	GO:0016072	GO BP	rRNA metabolic process	59	3.77	−15.85	−12.54
◯		GO:0061024	GO BP	membrane organization	119	7.61	−15.08	−11.82
	◯	GO:0006351	GO BP	DNA-templated transcription	95	6.08	−14.75	−11.53
	◯	R-MMU-199991	REAC	Membrane Trafficking	92	5.89	−13.88	−10.80
◯	◯	GO:0030705	GO BP	cytoskeleton-dependent intracellular transport	50	3.20	−13.72	−10.68
◯	◯	GO:0006886	GO BP	intracellular protein transport	92	5.89	−13.58	−10.63
◯	◯	GO:0006914	GO BP	autophagy	68	4.35	−13.57	−10.63
◯		GO:0048858	GO BP	cell projection morphogenesis	95	6.08	−13.12	−10.24
◯	◯	GO:0022411	GO BP	cellular component disassembly	58	3.71	−12.49	−9.67
◯	◯	R-MMU-6791226	REAC	Major pathway of rRNA processing in the nucleolus and cytosol	43	2.75	−11.76	−9.04
◯		GO:0070848	GO BP	response to growth factor	91	5.82	−10.63	−7.95
◯	◯	GO:0007626	GO BP	locomotory behavior	55	3.52	−10.60	−7.93
◯	◯	GO:0048193	GO BP	Golgi vesicle transport	53	3.39	−9.98	−7.34
◯		GO:1903829	GO BP	positive regulation of protein localization	84	5.37	−9.62	−7.02
	◯	GO:0034976	GO BP	response to endoplasmic reticulum stress	48	3.07	−9.60	−7.01
	◯	R-MMU-983799	REAC	Mitochondrial protein degradation	24	1.54	−9.29	−6.76
◯	◯	GO:0021955	GO BP	central nervous system neuron axonogenesis	18	1.15	−9.20	−6.68
◯	◯	R-MMU-6807505	REAC	RNA polymerase II transcribes snRNA genes	23	1.47	−9.17	−6.68

**Table 4 genes-17-00051-t004:** Cluster annotation associated with Appendix A.

Color	MCODE	GO	Description	Log10(P)
Red	MCODE_1	mmu05214	Glioma—Mus musculus (house mouse)	−7.8
Red	MCODE_1	WP1763	Mechanisms associated with pluripotency	−7.7
Red	MCODE_1	mmu05200	Pathways in cancer—Mus musculus (house mouse)	−7.2
Blue	MCODE_2	GO:0051123	RNA polymerase II preinitiation complex assembly	−11.7
Blue	MCODE_2	GO:0070897	transaction preinitiation complex assembly	−11.2
Blue	MCODE_2	GO:0060261	positive regulation of transcription initiation by RNA polymerase II	−10.8
Green	MCODE_3	GO:0032543	mitochondrial translation	−22.4
Green	MCODE_3	GO:0140053	mitochondrial gene expression	−20.8
Green	MCODE_3	R-MMU-5419276	Mitochondrial translation termination	−20.7
Purple	MCODE_4	GO:0042254	ribosome biogenesis	−19.5
Purple	MCODE_4	GO:0022613	ribonucleoprotein complex biogenesis	−17.6
Purple	MCODE_4	R-MMU-8868773	rRNA processing in the nucleus and cytosol	−17.5
Orange	MCODE_5	R-MMU-165159	MTOR signaling	−8.7
Orange	MCODE_5	GO:1903432	regulation of TORC1 signaling	−7.0
Orange	MCODE_5	R-MMU-166208	mTORC1-mediated signaling	−6.9
Yellow	MCODE_6	mmu03083	Polycomb repressive complex—Mus musculus (house mouse)	−9.9
Yellow	MCODE_6	R-MMU-8953750	Transcriptional Regulation by E2F6	−7.2
Yellow	MCODE_6	R-MMU-212436	Generic Transcription Pathway	−3.7
Brown	MCODE_7	R-MMU-983169	Class I MHC-mediated antigen processing & presentation	−5.7
Brown	MCODE_7	mmu04512	ECM–receptor interaction—Mus musculus (house mouse)	−5.6
Brown	MCODE_7	mmu04120	Ubiquitin-mediated proteolysis—Mus musculus (house mouse)	−4.9
Pink	MCODE_8	mmu04814	Motor proteins—Mus musculus (house mouse)	−12.2
Pink	MCODE_8	R-MMU-2132295	MHC class II antigen presentation	−10.5
Pink	MCODE_8	R-MMU-8856688	Golgi-to-ER retrograde transport	−10.4
Gray	MCODE_9	R-MMU-9013407	RHOH GTPase cycle	−10.7
Gray	MCODE_9	R-MMU-9012999	RHO GTPase cycle	−6.3
Gray	MCODE_9	R-MMU-194315	Signaling by Rho GTPases	−5.6
Teal	MCODE_10	GO:0036503	ERAD pathway	−8.7
Teal	MCODE_10	GO:0034976	response to endoplasmic reticulum stress	−7.1
Teal	MCODE_10	GO:0010498	proteasomal protein catabolic process	−5.8
Lavender	MCODE_12	R-MMU-9013424	RHOV GTPase cycle	−8.5
Lavender	MCODE_12	R-MMU-9012999	RHO GTPase cycle	−5.2
Lavender	MCODE_12	R-MMU-194315	Signaling by Rho GTPases	−4.7
Coral	MCODE_13	R-MMU-9006936	Signaling by TGFB family members	−6.9
Coral	MCODE_13	mmu04350	TGF-beta signaling pathway—Mus musculus (house mouse)	−6.9
Steel Blue	MCODE_14	R-MMU-6794361	Neurexins and neuroligins	−8.6
Steel Blue	MCODE_14	R-MMU-6794362	Protein–protein interactions at synapses	−7.6
Steel Blue	MCODE_14	GO:0034329	cell junction assembly	−5.5

**Table 5 genes-17-00051-t005:** Confirmation of Sst enrichment in the “Allen Brain Atlas 10x scRNA 2021” category using DNase-Seq predicted by AlphaGenome. ◯: Sst was in the top 10 ranked cell types and associated with adjusted *P*-values of less than 0.05. △: Sst was in top ranked 10 cell types but not associated with adjusted *P*-values of less than 0.05. ×: Sst was neither in top ranked 10 cell types nor associated with adjusted *P*-values of less than 0.05. $\ell_{1}=2$ only and $\ell_{1}=3$ only mean that common genes between $\ell_{1}=2$ and $\ell_{1}=3$ are excluded from $\ell_{1}=2$ and $\ell_{1}=3$, respectively.

Pairwise Comparisons			$\ell_{1}=2$	$\ell_{1}=3$	$\ell_{1}=2$ and $\ell_{1}=3$	$\ell_{1}=2$ Only	$\ell_{1}=3$ Only
Left cerebral cortex	>	Brain	△	△	◯	△	△
	<		◯	△	◯	◯	△
Right cerebral cortex	>		◯	△	◯	△	△
	<		◯	◯	△	◯	×
Layer of hippocampus	>		◯	△	◯	△	△
	<		◯	△	◯	◯	×
Frontal cortex	>		△	△	◯	△	△
	<		◯	△	△	△	×

## Data Availability

All data used in this study are available under GSE237419. Sample R code is available at https://github.com/tagtag/Sleep_deprivation/tree/main (accessed on 9 December 2025).

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
