# Peer review of "Somatostatin-Expressing Neurons Regulate Sleep Deprivation and Recovery"

_genes, 2026, doi:10.3390/genes17010051_

Round 1
Reviewer 1 Report
Comments and Suggestions for Authors
Review for Genes:
“Somatostatin-expressing neurons regulate sleep deprivation and recovery” by Kobayashi and Taguchi
Summary
Kobayashi and Taguchi used a public transcriptional profiling dataset (GSE237419 from Popescu et al. 2025) to examine the cortical gene expression profiles associated with sleep deprivation and recovery using tensor decomposition (TD)-based unsupervised feature extraction (FE) to re-analyze the data. The results from this decomposition are then explored using several functional enrichment-type analyses (Enrichr, AlphaGenome, MetaScape), and interpreted using a review of the literature detailing the role of somatostatin (SST) neurons in sleep regulation.
Major Comments
Kobayashi and Taguchi argue that their work is necessary because they are using a data-driven approach to examine patterns in the data instead of the “hypothesis-driven approach” used by Popescu et al. 2025. I suspect this premise is at least partially unfounded – most standard bioinformatics pipelines used for RNA-Seq include dimension reduction methods (e.g., principal components analysis (PCA) or multidimensional scaling (MDS)), with the identified principal components then compared to the variables of interest (and potential technical confounding variables) to identify general patterns in the data.
This does not rule out the possibility that their analysis using tensor decomposition-based unsupervised feature extraction provides some additional value, but this utility should be formally argued. For example, the authors could formally compare their results to those identified using more traditional methods (e.g., PCA) to demonstrate the added utility of their analysis procedure.
It would also be useful to demonstrate that the results produced by this method are valid – e.g., can be reproduced in other datasets examining a similar research question. Luckily, multiple public transcriptional profiling datasets exist that include cortical gene expression data from both sleep deprivation and recovery sleep conditions, so the authors should have access to the data needed to run a validation analysis (e.g. GSE33491 by Hinard et al 2012, GSE78215 by Gerstner et al. 2016, GSE144957 by Bjorness et al. 2020).
The RNA-Seq data analysis methods used by Kobayashi and Taguchi are very non-standard and often not fully described. For example, they appear to have read in a .txt file from the Gene Expression Omnibus website, which presumably contained a matrix of the summarized RNA-Seq counts for each gene for each sample. It is unclear where these values came from (presumably the analysis pipeline used by Popescu et al. 2025?), what the source was for the reference genome (genome build) and gene annotation, or whether the counts were normalized or log2 transformed (e.g., Log2CPM). Moreover, there is no mention of sample-level quality control, including screening for outlier samples or batch effects, or gene-level quality control (e.g., removing rows of data containing extremely low levels of expression or no variance). It is also unclear if their methods account for the non-linear mean-variance relationship that tends to exist in RNA-Seq data. Perhaps tensor decomposition-based unsupervised feature extraction is insensitive to these factors – if so, that should be explicitly stated.
Finally, the methods used to identify functional patterns in the results needs to be formally documented – especially since their article focuses heavily on the enrichment of the sleep deprivation/recovery gene expression signature within a set of genes enriched in somatostatin neurons (SST, as identified by Allen Brain Atlas). Currently, there is no description of the methods for MetaScape (discussed in the results). For Enrichr they have a single sentence saying that “The 100 top ranked genes were uploaded to Enrichr” which is incorrectly included in the methods section under AlphaGenome. The Metascape and Enrichr websites should be provided and properly cited, along with the actual methods/settings used. For example, for Enrichr it is particularly important to describe which gene set was used for the statistical background, as analyses using “whole genome” as the background often artificially identify brain-related gene sets as showing enrichment of differential expression (because the differential expression results are derived from brain tissue, and therefore only contain genes expressed in the brain). For interpreting false discovery rates (FDR), it is important to also know which of the *many* gene set libraries (and categories of gene set libraries) were referenced on the Enrichr website. Was Allen Brain Atlas the only gene set library used? Or just the only one out of 40 different gene set libraries that showed significant results?
Additional Minor Comments
Introduction: In order for the audience to interpret the results, it is important to overview the transcriptional profiling dataset used in the analysis in the introduction. For example, who were the experimental subjects and what were the different experimental conditions included in the experiment and why? How were those samples collected? For example, GSE237419 is cited as from Popescu et al. 2025, but the Gene Expression Omnibus record states that it includes samples from GSE140345 by Hor et al. 2019 and GSE113754 by Ingiosi et al. 2019, as well as novel samples. How were those different datasets harmonized?
Methods: Ideally, the analysis code should be released in association with the manuscript (e.g., a public Github repository). This is particularly true if one of the main contributions of the manuscript is the demonstration of a novel (or underused) analysis method.
Results: The extensive overview of the literature describing the tissue-specific role of SST-neurons in sleep regulation should be moved to the discussion section and greatly shortened to emphasize how the current study adds to the existing literature. Alternatively, the literature review could be mostly removed from the current paper and submitted as a short review paper or commentary.
Reviewer 2 Report
Comments and Suggestions for Authors
Dear authors,
The analytical workflow is clearly described, and the enrichment analysis is thorough. The use of tensor decomposition is an interesting methodological angle, and the manuscript is generally well written. However, the biological conclusions offered here are limited in novelty. The involvement of Sst expressing neurons in sleep regulation, particularly in slow wave activity and in the effects of sleep loss, is already well established in the literature, as reflected by the extensive referencing throughout the discussion. The study would benefit from clarifying what new insight is provided beyond the existing body of knowledge.
A major methodological concern arises from the decision to replace missing values in the data tensor with zeros (lines 34 to 40). This approach risks distorting the variance structure and artificially reinforcing condition specific patterns, potentially leading to biased or unstable latent components. Alternative imputation or modeling strategies should be considered or, at minimum, the implications of zero imputation should be explicitly discussed.
In addition, the gene lists obtained through the selection procedure are extremely large (>2000 genes), reducing interpretability and making specific biological conclusions difficult to substantiate. The procedure for σℓ1 optimization, although outlined (lines 47 to 62) , would benefit from further justification, since the current description remains opaque to readers who are not already familiar with the authors’ previous work.
The manuscript is further limited by the absence of experimental validation. While the authors argue that alpha genome predictions reinforce their claims (lines 255 to 273) , these in silico predictions cannot substitute for biological validation. Consequently, the assertion that Sst expressing neurons play a key regulatory role should be presented as a hypothesis generated from computational analysis rather than as a direct conclusion. Likewise, the extensive discussion of circuit mechanisms, including prefrontal projections to the lateral preoptic area and lateral hypothalamus (lines 185 to 240), is based entirely on external literature rather than on new data presented here. This material could be substantially reduced or reframed, since it currently gives the impression of over-interpretation.
Although Popescu et al., the original authors of the dataset, are referenced (lines 11 to 19) , no systematic comparison with their findings is performed. A direct comparison of gene signatures or pathway enrichments would strengthen the analysis and highlight the unique contribution, if any, of the tensor based approach. At present, the added value of the re-analysis remains unclear.
Overall, this study provides an interesting computational exploration of a sleep-related transcriptomic dataset and demonstrates that tensor decomposition can identify expression signatures related to known neuronal subtypes. However, because the key biological claims reproduce well-established knowledge, because missing-value handling raises concerns, and because no experimental validation is presented, the manuscript in its current form is not yet suitable. Substantial revision is required: the conclusions should be moderated, the novelty more clearly articulated, the methodological choices better justified, and the discussion refocused on what can genuinely be inferred from the analyses. With these changes, the study may offer a useful computational perspective on sleep-related transcriptional dynamics.
For these reason, I think this paper is valuable after minor revisions.
Best regards,
Reviewer 3 Report
Comments and Suggestions for Authors
The manuscript titled “Somatostatin-expressing neurons regulate sleep deprivation and recovery” has undergone a thorough evaluation. The review has identified several significant deficiencies, particularly regarding clarity and conciseness, which negatively impact the overall quality of the presentation.
The abstract is notably insufficient, comprising only a few lines. To enhance its effectiveness, it is essential to rewrite the abstract to include distinct components: a clearly defined objective outlining the study's purpose, a methods section detailing the experimental design and methodology, explicit results summarizing key findings, and a discussion and conclusion that contextualizes these findings within the broader academic discourse. This comprehensive approach will provide readers with a succinct and informative overview of the study's aims and outcomes.
Furthermore, the introduction requires expansion to provide an in-depth examination of the role of somatostatin-expressing neurons in regulating sleep deprivation. A thorough review of existing literature, accompanied by an analysis of the impact these neurons have on sleep and recovery mechanisms, is crucial for establishing a solid foundation for the study’s hypotheses and objectives. Such context will enable readers to appreciate the clinical relevance of the issues surrounding sleep deprivation and recovery.
Additionally, it is advisable to incorporate a well-structured flow diagram that visually represents the research methodology employed in this study. This diagram should systematically illustrate each step in the selection process for gene inclusion and exclusion criteria. A clear visual presentation will enhance transparency and aid readers in understanding the employed research design.
Concerning the statistical methods utilized, the manuscript requires a detailed overview of the statistical techniques applied throughout the research. Each statistical test should be explicitly justified, as this clarity will strengthen the credibility of the research and facilitate a comprehensive understanding of the data analysis processes.
Moreover, Figures 5 and 7 should be relocated to the supporting information section. These figures do not contribute substantially to the main text and are more appropriately placed as supplementary material.
Finally, the conclusion section necessitates a thorough revision to effectively communicate the broader implications of the findings. The authors should articulate the practical applications derived from their research and provide clear recommendations for future research directions. Such enhancements would significantly bolster the study's overall impact and relevance, ensuring it resonates within the academic community and beyond.
Reviewer 4 Report
Comments and Suggestions for Authors
The papers may present a very interesting topic and important conclusions. However, when it comes to writing the methodology, there are significant gaps that must be addressed.
The abstract is too short and jumps too quickly and directly into the contributions and results. I recommend rewriting it to about 200–250 words following the IMRaD structure (Introduction, Methods, Results, and Discussion) as one paragraph. In this revised version, the problem statement, research gaps, and proposed solutions should be clearly illustrated.
The introduction section needs to be expanded to about 2.5–3 pages, providing a detailed discussion of related works, research gaps, and the paper’s main contributions. This section should also include a clear statement of motivation and the research problem at the beginning.
After reviewing related works, a summary table can be included to clearly illustrate the contributions of existing studies and the identified research gaps.
The sections of the paper are currently too brief; please expand them. At the beginning of each section, include an introductory paragraph outlining its purpose and subsections.
Expand and enlarge the results and data (curves, charts, tables, etc) description in details.
Please revise your paper thoroughly in light of these points and add appropriate references to related works to ensure a complete and methodologically sound paper. These comments are important to strengthen both the methodological rigor and the informative value of your work.
Round 2
Reviewer 1 Report
Comments and Suggestions for Authors
The authors did not address my original concerns with the paper. The value of their paper is rooted in the idea that they are re-analyzing the data from a previous paper using a superior method, but they were not able to validate their results. They appear to be working with gene expression data without performing standard quality control or variance stabilization procedures, using a partially processed RNA-Seq count matrix downloaded from GEO. They are unwilling to describe the basic aspects of their methodology - for example, they refuse to provide the settings, statistical background, or number of gene set libraries referenced for Enrichr. This means that they are likely to have used default settings (background as whole genome) - which suggests that their functional ontology results - i.e., the main findings that they emphasize - are probably just artifact.
Reviewer 3 Report
Comments and Suggestions for Authors
The authors made significant revisions based on the reviewers' comments; therefore, the manuscript is acceptable for publication.
Reviewer 4 Report
Comments and Suggestions for Authors
The authors have done their best to address my comments, and the paper now appears improved. However, there is still an issue with the figures: please do not wrap or distort them. When adjusting their size, ensure that the original proportions are preserved.
